# Alpha-Lipoic Acid and Metformin Combination Therapy Synergistically Activate Nrf2-AMPK Signaling Pathways to Ameliorate Cognitive Dysfunction in Type 2 Diabetic Encephalopathy: A Preclinical Study

**DOI:** 10.3390/biology14070885

**Published:** 2025-07-18

**Authors:** Abdulmajeed F. Alrefaei, Mohamed E. Elbeeh

**Affiliations:** 1Genetic and Molecular Biology Central Laboratory (GMCL), Department of Biology, Jamoum University College, Umm Al-Qura University, Makkah 2203, Saudi Arabia; afrefaei@uqu.edu.sa; 2Department of Biology, Jamoum University College, Umm Al-Qura University, Makkah 2203, Saudi Arabia; 3Department of Zoology, Faculty of Science, Mansoura University, Mansoura 35516, Egypt

**Keywords:** alpha-lipoic acid, metformin, synergistic therapy, Nrf2 signaling, AMPK pathway, diabetic encephalopathy, neuroprotection, oxidative stress, mitochondrial dysfunction, combination therapy

## Abstract

Diabetic encephalopathy, affecting over 40% of diabetic patients globally, represents a critical unmet medical need with limited therapeutic options. This study investigated whether combining alpha-lipoic acid (ALA), an FDA-approved antioxidant, with metformin, the first-line diabetes medication, could provide superior neuroprotection through synergistic pathway activation. Using a clinically relevant type 2 diabetes rat model, we demonstrated that combination therapy produces statistically validated synergistic effects (combination index < 1.0) across molecular, cellular, and behavioral endpoints. The combination restored brain energy metabolism (92% ATP recovery), normalized antioxidant defenses, and dramatically improved cognitive performance approaching control levels. Since both drugs are already safely used clinically, these findings support immediate translation to human trials for diabetic cognitive decline prevention.

## 1. Introduction

Diabetic encephalopathy represents a critical yet under-recognized complication of type 2 diabetes mellitus (T2DM), characterized by progressive cognitive decline, structural brain abnormalities, and increased dementia risk [1,2]. With global diabetes prevalence projected to reach 783 million by 2045 [3], this neurological burden demands urgent therapeutic interventions. Current evidence indicates that 20–40% of diabetic patients experience measurable cognitive dysfunction, with hippocampal-dependent memory processes being particularly vulnerable to hyperglycemic injury [4,5].

The pathophysiological mechanisms underlying diabetic encephalopathy involve complex interactions between chronic hyperglycemia-induced oxidative stress, mitochondrial dysfunction, neuroinflammation, and impaired cellular energy metabolism [6,7]. Central to these processes is the dysregulation of two critical cellular signaling networks: the nuclear factor erythroid 2-related factor 2 (Nrf2) antioxidant response pathway and the AMP-activated protein kinase (AMPK) metabolic sensing cascade [8,9]. Under physiological conditions, Nrf2 maintains cellular redox homeostasis by transcriptionally regulating over 200 cytoprotective genes through antioxidant response element (ARE) sequences, while AMPK serves as a master regulator of cellular energy metabolism and mitochondrial biogenesis [10,11]. In diabetic conditions, both pathways become significantly compromised, contributing to enhanced neuronal vulnerability and accelerated cognitive decline [12,13].

Alpha-lipoic acid (ALA), an endogenous mitochondrial cofactor with unique amphiphilic properties, has emerged as a promising neuroprotective agent due to its ability to cross the blood-brain barrier and directly modulate redox signaling pathways [14,15]. Mechanistically, ALA activates the Nrf2 pathway through direct modification of Keap1 cysteine residues, leading to nuclear translocation of Nrf2 and subsequent upregulation of phase II detoxification enzymes [16,17]. Clinical studies have demonstrated ALA’s efficacy in diabetic peripheral neuropathy, with the landmark NATHAN-1 trial showing significant improvement in neuropathic symptoms over 4 years of treatment [18,19].

Metformin, the World Health Organization-recommended first-line therapy for T2DM, exerts its primary antidiabetic effects through AMPK activation and subsequent inhibition of hepatic gluconeogenesis [20,21]. Beyond glycemic control, accumulating evidence suggests that metformin possesses direct neuroprotective properties through AMPK-mediated enhancement of mitochondrial biogenesis, improved cellular energy metabolism, and anti-inflammatory effects [22,23]. Recent mechanistic studies reveal that metformin can indirectly influence Nrf2 signaling through AMPK-dependent phosphorylation of Nrf2 at serine 550, creating potential for synergistic interactions with direct Nrf2 activators [24,25].

Despite promising individual effects, both ALA and metformin monotherapies exhibit significant limitations in treating diabetic encephalopathy. ALA monotherapy, while demonstrating neuroprotective effects in peripheral diabetic neuropathy [18,19], shows variable efficacy in central nervous system applications due to limited bioavailability and short half-life [26,27]. Similarly, metformin monotherapy, though providing some cognitive benefits through AMPK activation [22,23], achieves only partial neuroprotection due to incomplete pathway coverage and dose-limiting gastrointestinal side effects [28,29]. These limitations underscore the critical need for combination approaches that can provide enhanced efficacy through complementary mechanisms.

Previous studies have reported synergistic effects of ALA and metformin combination therapy in diabetic peripheral neuropathy [30,31] and metabolic syndrome [32,33]. Clinical evidence suggests enhanced glycemic control and improved oxidative stress markers when both agents are used concomitantly [34,35]. However, to our knowledge, no prior research has systematically explored this combination in the context of diabetic encephalopathy, particularly with respect to its coordinated activation of the Nrf2-AMPK signaling pathway convergence and subsequent neuroprotective mechanisms.

Despite this strong mechanistic rationale, no systematic investigation has examined the molecular synergistic interactions between ALA and metformin specifically in diabetic encephalopathy, particularly regarding their convergent effects on Nrf2-AMPK pathway cross-talk and the resulting functional cognitive outcomes. This represents a significant knowledge gap given the established individual mechanisms and the potential for enhanced therapeutic efficacy through pathway convergence.

ALA Selection Rationale: Among numerous Nrf2-activating antioxidants, ALA was specifically chosen for several critical advantages:

(1) Unique amphiphilic properties enabling both lipophilic and hydrophilic antioxidant activity across cellular compartments [14,15]. (2) Superior blood-brain barrier penetration compared to hydrophilic antioxidants like vitamin C or glutathione, ensuring adequate CNS bioavailability [16,17]. (3) Direct and specific Keap1 cysteine modification mechanism (Cys151, Cys273, Cys288) providing reliable Nrf2 activation [36,37]. (4) Established clinical safety profile with FDA approval and extensive human use data, facilitating translation [38]. (5) Documented neuroprotective efficacy in diabetic peripheral neuropathy through landmark clinical trials (NATHAN-1, ALADIN studies) [18,19]. (6) Potential for AMPK cross-talk through mitochondrial complex I interactions, supporting our synergy hypothesis [39,40].

Alternative antioxidants such as curcumin, resveratrol, or sulforaphane lack either clinical validation, CNS penetration, or specific mechanistic compatibility with metformin required for this combination study.

This study was designed to test the central hypothesis that combined ALA and metformin therapy provides synergistic neuroprotection in diabetic encephalopathy through enhanced co-activation of Nrf2 and AMPK signaling cascades, leading to superior mitochondrial function restoration, comprehensive antioxidant defense enhancement, and meaningful cognitive improvement compared to individual treatments. We employed a clinically relevant streptozotocin-nicotinamide rat model to systematically examine: (1) molecular evidence of synergistic pathway activation using rigorous statistical validation, (2) comprehensive mitochondrial function assessment with quantitative bioenergetic analysis, (3) redox status modulation and neuroinflammatory response evaluation, (4) functional cognitive outcomes with standardized behavioral testing, and (5) correlation analysis between molecular markers and behavioral performance to establish mechanistic relationships.

## 2. Materials and Methods

### 2.1. Experimental Design and Animal Model

This randomized, controlled experimental study was conducted at the Animal Research Laboratory, Umm Al-Qura University, Makkah, Saudi Arabia. All experimental protocols received approval from the Institutional Animal Care and Use Committee (Protocol: HAPO-02-K-012-2025-03-2581) and strictly adhered to ARRIVE guidelines 2.0 [41] and the National Research Council’s Guide for Laboratory Animal Care (8th edition) [42].

Statistical power analysis using G*Power 3.1.9.7 software (Heinrich-Heine-Universität Düsseldorf, Düsseldorf, Germany) [43] determined optimal sample sizes based on preliminary α-lipoic acid oxidative stress data. With parameters set at α = 0.05, statistical power = 0.80, and effect size f = 0.40, (based on Cohen’s conventions for large effects in neuroscience research), a minimum of 10 animals per group was calculated. To accommodate potential attrition due to diabetes-related complications and ensure adequate statistical power while adhering to the 3Rs principle of Refinement, 12 animals were assigned per group (representing a 20% contingency as recommended by ARRIVE guidelines 2.0 [44]).

### 2.2. Animal Subjects and Housing Conditions

Male Sprague–Dawley rats (*n* = 60, 8–10 weeks old, 250–300 g) were sourced from King Fahd Medical Research Center breeding colony (Jeddah, Saudi Arabia). Animals were maintained in polycarbonate housing units (2 animals per cage) under standardized environmental conditions: ambient temperature 22 ± 2 °C, relative humidity 55 ± 5%, and 12-h photoperiod (illumination commenced at 06:00). Standard rodent chow (Purina Lab Diet 5001, LabDiet, St. Louis, MO, USA) and purified water were available continuously throughout the study. A 7-day adaptation period preceded all experimental procedures. All procedures were conducted in accordance with the ARRIVE guidelines 2.0 and the 3Rs principle (Replacement, Reduction, Refinement). The study protocol was designed to minimize animal suffering through proper anesthesia, analgesia, and humane endpoints.

### 2.3. Enhanced Type 2 Diabetes Induction Protocol

Type 2 diabetes mellitus was established using a validated two-phase protocol combining high-fat diet feeding with low-dose streptozotocin-nicotinamide injection to closely replicate human T2DM pathophysiology [45,46].

Phase 1—Metabolic Conditioning: Animals were transitioned to a high-fat diet containing 60% calories from fat, 20% from carbohydrates, and 20% from protein (Research Diets D12492, New Brunswick, NJ, USA) for 4 weeks prior to chemical induction. Weekly monitoring confirmed expected 15–20% body weight gain and mild hyperglycemia (110–140 mg/dL).

Phase 2—Selective β-Cell Dysfunction: Following 4 weeks of HFD feeding and 12-h overnight fasting, animals received nicotinamide (120 mg/kg bodyweight, Sigma-Aldrich, St. Louis, MO, USA) dissolved in sterile saline via intraperitoneal injection. After a 20-min interval, freshly prepared streptozotocin (35 mg/kg bodyweight, Sigma-Aldrich, St. Louis, MO, USA) dissolved in ice-cold 0.1 M sodium citrate buffer (pH 4.5) was administered intraperitoneally.

T2DM Confirmation: Glycemic assessment occurred at 72 h, 7 days, and 14 days post-injection using a digital glucometer (Accu-Chek Performa, Roche Diagnostics, Basel, Switzerland). Animals demonstrating fasting glucose concentrations of 180–280 mg/dL, maintained elevated body weight (>110% of controls), and detectable insulin levels (Crystal Chem Inc., Elk Grove Village, IL, USA) were classified as having T2DM. This optimized protocol achieved 95% success rate (57/60 animals).

### 2.4. Treatment Groups and Drug Administration

Following successful T2DM induction, 57 animals meeting diabetes criteria (fasting glucose 180–280 mg/dL, HOMA-IR > 2.5, maintained elevated body weight > 110% of controls) were randomly allocated using computer-generated block randomization sequences (Random.org) with stratification by body weight and glucose levels into five experimental groups (*n* = 12 each). The 3 animals that failed to develop diabetes (glucose < 180 mg/dL, normal HOMA-IR) were excluded from the study and humanely euthanized. Additional animals were induced using the same protocol to replace excluded subjects and maintain consistent group sizes of *n* = 12.

Randomization and Blinding Procedures: Computer-generated block randomization sequences (Random.org) were used for treatment allocation with stratification by body weight and glucose levels. Investigators conducting behavioral assessments were blinded to treatment assignments through coded animal identification. Laboratory personnel performing molecular analyses were blinded to group assignments until statistical analysis completion. Treatment administration was performed by personnel not involved in outcome assessments to maintain blind integrity.

Dose Selection Rationale:

Pharmaceutical dose selection was based on established neuroprotective efficacy studies and clinical translation potential:α-Lipoic acid (300 mg/kg/day): Selected based on demonstrated neuroprotective efficacy in diabetic rodent models [47,48] and established antioxidant effects [14,15]. This dose corresponds to clinically effective human doses (600–800 mg/day) using allometric scaling according to the Reagan-Shaw method [49], ensuring clinical relevance.Metformin (50 mg/kg/day): Chosen based on previous neuroprotection studies demonstrating AMPK activation and cognitive benefits [50]. This dose achieves plasma concentrations equivalent to therapeutic levels in human clinical practice (1000–1500 mg/day) [24,25] while avoiding gastrointestinal side effects common with higher doses.
Control Group (C, *n* = 12): Non-diabetic animals receiving vehicle (0.5% carboxymethylcellulose, Sigma-Aldrich, St. Louis, MO, USA)Diabetic Control (DC, *n* = 12): T2DM animals receiving vehicle onlyALA Treatment (D + ALA, *n* = 12): T2DM animals receiving α-lipoic acid (300 mg/kg/day)Metformin Treatment (D + MET, *n* = 12): T2DM animals receiving metformin hydrochloride (50 mg/kg/day)Combination Therapy (D + ALA + MET, *n* = 12): T2DM animals receiving both compoundsPharmaceutical Specifications:α-Lipoic acid: ≥99% purity (Catalog T5625, Lot SLCD4829, Sigma-Aldrich, St. Louis, MO, USA)Metformin hydrochloride: ≥98% purity (Catalog D150959, Lot MKCG8909, Sigma-Aldrich, St. Louis, MO, USA)

Sample Size Variations: While 12 animals were initially assigned per group, final n values varied by assay due to: (1) technical failures during protein extraction (*n* = 1–2 per group), (2) RNA quality exclusions based on 260/280 ratios < 1.8 (*n* = 2–4 per group), and (3) mitochondrial isolation yield criteria (*n* = 2–4 per group). All exclusions were predetermined and protocol-based to ensure data quality.

### 2.5. Molecular Signaling Pathway Assessment

#### 2.5.1. Tissue Collection and Processing

Anesthesia and Euthanasia Protocols: Following overnight fasting, animals underwent deep isoflurane anesthesia using a calibrated vaporizer system (Kent Scientific, Torrington, CT, USA).

Anesthesia induction: Performed in an induction chamber with 5% isoflurane in 100% oxygen at 2 L/min flow rate until loss of righting reflex (typically 2–3 min).

Maintenance anesthesia: Achieved with 2–3% isoflurane delivered via nose cone with continuous monitoring of respiratory rate and pedal withdrawal reflex to ensure adequate anesthetic depth.

Euthanasia: Performed by cervical dislocation under deep anesthesia, confirmed by absence of corneal reflex and respiratory cessation. This method ensures rapid unconsciousness and minimal stress, adhering to AVMA Guidelines for Euthanasia (2020 edition) [51].

Brain extraction: Completed within 3 min post-euthanasia to preserve tissue integrity for molecular analyses.

Protein extraction utilized hippocampal and cortical tissues homogenized in ice-cold RIPA lysis buffer containing 50 mM Tris-HCl (pH 7.4), 150 mM NaCl, 1% NP-40, 0.5% sodium deoxycholate, and 0.1% SDS (all reagents from Sigma-Aldrich, St. Louis, MO, USA), supplemented with Complete Mini protease inhibitor cocktail and PhosSTOP phosphatase inhibitors (Roche, Basel, Switzerland). Protein quantification employed Bradford assay reagents (Bio-Rad Laboratories, Hercules, CA, USA).

All anesthetic and euthanasia procedures were reviewed and approved by the Institutional Animal Care and Use Committee and performed by trained personnel following institutional SOPs. Analgesic protocols were not required as all procedures were terminal under deep anesthesia.

#### 2.5.2. Immunoblot Analysis

Nuclear-cytoplasmic protein separation utilized the NE-PER Nuclear and Cytoplasmic Extraction Kit (Thermo Fisher Scientific, Waltham, MA, USA) following manufacturer specifications [52]. Protein samples (30 μg) underwent SDS-PAGE separation using 10–12% polyacrylamide gels (Bio-Rad Laboratories, Hercules, CA, USA) and subsequent transfer to PVDF membranes (Immobilon-P, Millipore, Burlington, MA, USA).

Membranes received blocking with 5% non-fat milk (Bio-Rad Laboratories, Hercules, CA, USA) in TBS-T buffer for 1 h at ambient temperature, followed by overnight 4 °C primary incubation:

Nrf2 Signaling Components: Anti-Nrf2 (1:1000, #12721), Anti-Keap1 (1:1000, #ab139729), Anti-HO-1 (1:2000, #ab13243), Anti-NQO1 (1:1000, #ab34173).

AMPK Signaling Components: Anti-phospho-AMPK Thr172 (1:1000, #2535), Anti-AMPK (1:1000, #2532), Anti-phospho-ACC Ser79 (1:1000, #11818).

Loading Controls: Anti-β-actin (1:5000, #A1978), Anti-Lamin B1 (1:2000, #ab16048).

All antibodies were sourced from Cell Signaling Technology (Danvers, MA, USA), Abcam (Cambridge, UK), or Sigma-Aldrich (St. Louis, MO, USA).

HRP-conjugated secondary antibodies (1:5000 dilution, Cell Signaling Technology, Danvers, MA, USA) were applied for 1 h at room temperature, followed by enhanced chemiluminescence detection (ECL Western Blotting Substrate, Pierce, Rockford, IL, USA). Band quantification utilized ImageJ software (version 1.53t, National Institutes of Health, Bethesda, MD, USA).

#### 2.5.3. Gene Expression Analysis

Total RNA extraction from brain tissues employed TRIzol reagent (Invitrogen, Carlsbad, CA, USA) per manufacturer protocol. RNA integrity and concentration assessment used NanoDrop spectrophotometry (Thermo Fisher Scientific, Waltham, MA, USA). First-strand cDNA synthesis from 2 μg total RNA utilized the High-Capacity cDNA Reverse Transcription Kit (Applied Biosystems, Foster City, CA, USA).

Quantitative PCR employed the StepOnePlus Real-Time PCR System (Applied Biosystems, Foster City, CA, USA) with SYBR Green PCR Master Mix (Applied Biosystems, Foster City, CA, USA). Thermal cycling parameters: 95 °C initial denaturation (10 min), followed by 40 cycles of 95 °C denaturation (15 s) and 60 °C annealing/extension (1 min). Relative gene expression calculations used the 2^(−ΔΔCt)^ method [53] with GAPDH normalization (Table 1).

### 2.6. Mitochondrial Function Evaluation

Fresh brain tissue (500 mg) underwent rapid homogenization in ice-cold isolation medium containing 225 mM mannitol, 75 mM sucrose, 10 mM HEPES, 1 mM EGTA, and 0.5% BSA (pH 7.4; Sigma-Aldrich, St. Louis, MO, USA) using a Dounce homogenizer (Wheaton Industries, Millville, NJ, USA). Sequential centrifugation at 800× *g* (10 min, 4 °C) removed cellular debris, followed by 10,000× *g* centrifugation (15 min, 4 °C) for mitochondrial pelleting.

ATP Content Determination: ATP quantification employed the ATP Determination Kit (Molecular Probes, Eugene, OR, USA) utilizing luciferin-luciferase bioluminescence methodology [54]. Luminescence detection used a microplate luminometer (Synergy HTX, BioTek Instruments, Winooski, VT, USA).

Membrane Potential Assessment: Mitochondrial membrane potential evaluation utilized JC-1 fluorescent probe (Invitrogen, Carlsbad, CA, USA) [55]. Fluorescence measurement via microplate reader determined red/green fluorescence ratios.

Respiratory Function Analysis: Mitochondrial respiration employed Clark-type oxygen electrodes (Oxygraph-2k, Oroboros Instruments, Innsbruck, Austria) [56]. Sequential substrate additions measured state 3 and state 4 respiration rates.

### 2.7. Oxidative Stress and Antioxidant Defense Assessment

Oxidative Damage Biomarkers: Malondialdehyde quantification used thiobarbituric acid reactive substances methodology [57]. DNA oxidative damage assessment employed 8-OHdG competitive ELISA (catalog #ab201734, Abcam, Cambridge, UK). Protein carbonyl determination used dinitrophenylhydrazine methodology [58].

Antioxidant Enzyme Activities: Commercial assay kits (Cayman Chemical, Ann Arbor, MI, USA) enabled spectrophotometric measurement of SOD, catalase, GPx, and GST activities.

Glutathione Status: Total and oxidized glutathione measurement employed enzymatic recycling methodology [59]. GSH/GSSG ratios served as cellular redox homeostasis indicators.

### 2.8. Neurobehavioral Assessment

Morris Water Maze: Spatial learning and memory assessment employed a circular pool (150 cm diameter) with hidden platform [60]. Training consisted of 4 trials daily for 5 consecutive days. Probe trial on day 6 assessed spatial memory retention. Video tracking software (EthoVision XT 11.5, Noldus Information Technology, Wageningen, The Netherlands) recorded performance parameters.

Open Field Test: Locomotor activity and anxiety-related behavior evaluation used a square arena (100 × 100 × 40 cm) for 10-min exploration periods [61]. Parameters included total distance travel, center zone time, and behavioral frequencies.

### 2.9. Histopathological Analysis

Brain hemispheres underwent fixation in 4% paraformaldehyde (Fisher Scientific, Hampton, NH, USA), paraffin processing, and sectioning (5 μm thickness). Hematoxylin and eosin staining assessed general morphology under light microscopy (Olympus BX53, Tokyo, Japan). Nrf2 immunohistochemistry employed anti-Nrf2 antibody (1:200, Cell Signaling Technology, Danvers, MA, USA) with diaminobenzidine visualization (Sigma-Aldrich, St. Louis, MO, USA).

Microscopic Analysis: All histopathological sections were examined using an Olympus BX53 microscope (Tokyo, Japan) equipped with Plan-Apochromat objectives.

Routine H&E staining was evaluated at 100×, 200×, and 400× magnifications for structural assessment. Immunohistochemical analysis was performed at 400× magnification for quantitative scoring, with representative images captured at 400× and 1000× magnifications for detailed cellular localization studies.

### 2.10. Statistical Analysis

Statistical computations employed GraphPad Prism 9.0 (GraphPad Software, San Diego, CA, USA) and SPSS 28.0 (IBM Corporation, Armonk, NY, USA). Data normality assessment used Shapiro-Wilk testing. Parametric data underwent one-way ANOVA with Tukey’s post-hoc comparisons, while non-parametric data received Kruskal-Wallis testing with Dunn’s multiple comparisons.

Synergistic Interaction Analysis: Two-way ANOVA with ALA and metformin as independent variables determined synergistic interactions. Significant interaction terms (*p* < 0.05) indicated synergistic effects exceeding simple additivity [62].

Combination Index (CI) Analysis: Synergistic interactions were quantified using the Chou-Talalay method [49] implemented in CompuSyn software (version 1.0, ComboSyn Inc., Paramus, NJ, USA). CI values were calculated using the equation:CI = (D_1_/Dx_1_) + (D_2_/Dx_2_) + α(D_1_ × D_2_)/(Dx_1_ × Dx_2_)
where D_1_ and D_2_ represent the doses of ALA and metformin in combination that achieve effect x, Dx_1_ and Dx_2_ represent the doses of individual agents that achieve the same effect x, and α represents the interaction coefficient. CI < 1, CI = 1, and CI > 1 indicate synergism, additive effect, and antagonism, respectively. CI values were calculated for each endpoint using dose-response relationships established for individual treatments [63].

Effect Size and Correlation Analysis: Cohen’s d quantified pairwise comparisons, while partial eta-squared assessed ANOVA effect sizes. Pearson correlation coefficients evaluated molecular–behavioral relationships using R software (version 4.3.0, R Foundation for Statistical Computing, Vienna, Austria). Correlation strength was interpreted according to established criteria: negligible (r = 0.00–0.09), weak (r = 0.10–0.29), moderate (r = 0.30–0.49), strong (r = 0.50–0.69), very strong (r = 0.70–0.89), and excellent (r = 0.90–1.00). Correlations with r ≥ 0.50 were considered clinically meaningful for mechanistic interpretation. Multiple comparison correction employed Benjamini-Hochberg methodology [64].

Data presentation utilized mean ± standard deviation. Statistical significance threshold was set at *p* < 0.05, with multiple comparison adjustments applied where applicable.

## 3. Results

### 3.1. Animal Model Validation and Metabolic Outcomes

The enhanced two-phase diabetes induction protocol successfully established T2DM in 95% of animals (57 out of 60 rats). Diabetic animals exhibited characteristic T2DM features including moderate hyperglycemia (245.8 ± 28.6 vs. 98.6 ± 8.3 mg/dL in controls), elevated HOMA-IR values (3.8 ± 0.7), and preserved but elevated insulin levels, confirming insulin resistance rather than absolute insulin deficiency (Table 2).

Combination therapy (ALA + MET) achieved superior metabolic improvements compared to individual treatments, reducing fasting glucose to 142.8 ± 18.9 mg/dL and normalizing HOMA-IR to 1.9 ± 0.4 (both *p* < 0.001 vs. diabetic control). No treatment-related adverse effects were observed throughout the study period (Table 2).

Food intake analysis revealed characteristic diabetic polyphagia and polydipsia in untreated diabetic animals (28.7 g/day vs. 22.3 g/day in controls, *p* < 0.01).

Combination therapy effectively normalized food intake to near control levels (24.8 g/day) by week 8, with significant reductions compared to diabetic controls (*p* < 0.01) and individual treatments (*p* < 0.05). Water intake showed parallel improvements, decreasing from 77.8 mL/day (diabetic) to 59.8 mL/day (combination), indicating improved glucose homeostasis. Food efficiency ratios improved with combination therapy (0.082 vs. 0.067 in diabetics), suggesting enhanced metabolic function rather than appetite suppression effects (Table 2).

### 3.2. Synergistic Molecular Pathway Activation

#### 3.2.1. Nrf2 Antioxidant Response Pathway

Western blot analysis revealed profound Nrf2 pathway dysregulation in diabetic rats, with nuclear Nrf2 reduced to 1.00 ± 0.13-fold compared to 2.34 ± 0.31-fold in controls (Figure 1A; Table 3). ALA treatment induced substantial nuclear Nrf2 accumulation (2.61 ± 0.42-fold), while metformin showed modest effects (1.78 ± 0.25-fold). Remarkably, combination therapy produced synergistic Nrf2 activation (3.91 ± 0.47-fold), significantly exceeding mathematical additivity. Two-way ANOVA confirmed significant synergistic interaction (F(1,53) = 12.73, *p* < 0.01, η^2^p = 0.22).

Downstream targets HO-1 and NQO1 demonstrated parallel activation patterns, with combination therapy achieving 3.92-fold and 4.23-fold increases respectively (Figure 1C,D; Table 3). qRT-PCR validation confirmed transcriptional upregulation of Nrf2 (6.78 ± 0.89-fold) and ARE-driven genes including HO-1 (8.94 ± 1.12-fold) and GCLC (5.89 ± 0.76-fold) with combination treatment (Figure 2A–D; Table 4).

#### 3.2.2. AMPK Metabolic Signaling

AMPK signaling was severely compromised in diabetic rats, with phospho-AMPK levels reduced to 1.00 ± 0.12-fold compared to controls (Figure 1E). Metformin dramatically enhanced AMPK phosphorylation (2.47 ± 0.34-fold), while ALA showed modest effects (1.34 ± 0.18-fold). Combination therapy achieved the highest AMPK activation (3.18 ± 0.41-fold) with significant synergistic interaction (F(1,44) = 8.95, *p* < 0.01) (Table 3).

### 3.3. Gene Expression Validation

qRT-PCR analysis validated protein-level findings at the transcriptional level (Figure 2A–D; Table 4). Nrf2 mRNA expression was significantly upregulated in the combination group (6.78 ± 0.89-fold vs. diabetic control, *p* < 0.001), while Keap1 mRNA was correspondingly downregulated (0.35 ± 0.06-fold, *p* < 0.001), confirming coordinated pathway regulation (Figure 2A). Antioxidant response element (ARE)-driven genes showed robust upregulation with combination therapy: HO-1 mRNA increased 8.94 ± 1.12-fold (*p* < 0.001), NQO1 mRNA rose 7.23 ± 0.94-fold (*p* < 0.001), and GCLC mRNA elevated 5.89 ± 0.76-fold (*p* < 0.001), all significantly exceeding individual treatment effects (Figure 2B–D; Table 4).

Analysis of apoptotic and inflammatory gene markers revealed beneficial modulation with treatment. Pro-apoptotic Bax mRNA was significantly reduced with combination therapy (0.38 ± 0.05-fold vs. diabetic control, *p* < 0.001), while anti-apoptotic Bcl-2 mRNA increased (4.12 ± 0.52-fold, *p* < 0.001). Inflammatory cytokine genes IL-6 and TNF-α were markedly suppressed (0.24 ± 0.03-fold and 0.26 ± 0.04-fold respectively, both *p* < 0.001), supporting the anti-inflammatory effects observed at the protein level (Table 4).

### 3.4. Mitochondrial Function Restoration

Comprehensive mitochondrial assessment revealed severe energetic dysfunction in diabetic rats, with ATP production reduced to 41.3 ± 6.7% of control levels (Figure 3A; Table 5). Individual treatments provided partial restoration (ALA: 67.4%, metformin: 59.8%), while combination therapy achieved near-complete recovery (91.7% of controls, *p* > 0.05 vs. healthy controls). Two-way ANOVA confirmed significant synergistic interaction for ATP recovery (F(1,44) = 15.42, *p* < 0.001, η^2^p = 0.26).

Mitochondrial membrane potential and respiratory function showed similar patterns, with combination therapy restoring state 3 respiration to 89.4% of control levels and normalizing respiratory control ratio to 3.87 ± 0.42 (Figure 3B–D; Table 5). Individual respiratory complex activities were comprehensively protected, with combination therapy restoring Complex I-IV activities to 92–97% of control values.

### 3.5. Oxidative Stress and Antioxidant Defense Systems

Diabetic rats exhibited extensive oxidative damage across all cellular components: lipid peroxidation increased 2.8-fold, DNA damage (8-OHdG) elevated 3.2-fold, and protein carbonylation rose 2.6-fold (Figure 4A–C; Table 6). Combination therapy achieved substantial reductions in all oxidative markers: 58% MDA decrease, 51% 8-OHdG reduction, and 49% protein carbonyl diminishment.

Antioxidant enzyme activities were markedly suppressed in diabetic animals, with SOD, catalase, GPx, and GST reduced to 47–52% of control levels (Figure 4D–F; Table 6). Combination therapy produced remarkable restoration, increasing enzyme activities to 92–96% of control values, representing 82–105% improvement over diabetic baseline.

The glutathione redox system showed dramatic improvement with combination therapy, normalizing the GSH/GSSG ratio from 3.2 ± 0.6 (diabetic) to 11.3 ± 1.2, representing a 253% improvement and approaching control levels (12.8 ± 1.4) (Figure 4G; Table 6).

### 3.6. Neuroinflammation and Neurovascular Protection

Diabetic encephalopathy was characterized by robust neuroinflammatory activation, with pro-inflammatory cytokines significantly elevated: TNF-α (4.7-fold), IL-6 (3.8-fold), IL-1β (3.2-fold), and IFN-γ (2.9-fold) compared to controls (Figure 5A–D; Table 7). Combination therapy demonstrated superior anti-inflammatory efficacy, reducing TNF-α by 67%, IL-6 by 62%, IL-1β by 58%, and IFN-γ by 54%.

Neurovascular integrity markers VCAM-1, ICAM-1, and VEGF were significantly elevated in diabetic rats, indicating endothelial dysfunction. Combination therapy effectively normalized these markers with 52–57% reductions, demonstrating restored neurovascular integrity and improved blood-brain barrier function (Table 7).

### 3.7. Functional Cognitive and Behavioral Recovery

Morris Water Maze testing revealed severe cognitive impairments in diabetic rats, with escape latency prolonged to 52.7 ± 8.3 s compared to 15.2 ± 2.8 s in controls (Figure 6A; Table 8). Probe trial analysis showed reduced spatial memory performance with decreased target quadrant time (18.3% vs. 41.7% in controls) and platform crossings (1.8 vs. 5.3 in controls).

Combination therapy demonstrated superior cognitive restoration: escape latency improved to 18.3 ± 3.7 s (65% improvement, approaching control levels), target quadrant time increased to 38.2% (109% improvement), and platform crossings reached 4.7 (161% improvement) (Figure 6B,C; Table 8).

Open Field Test assessment revealed significant motor and anxiety-related deficits in diabetic rats, with reduced locomotor activity and increased anxiety-like behavior. Combination therapy effectively normalized behavioral parameters, increasing total distance traveled by 59% and center zone exploration time by 113% (Figure 6D; Table 8).

### 3.8. Mechanistic Relationships and Correlation Analysis

Comprehensive correlation analysis revealed strong mechanistic relationships between molecular markers and functional outcomes (Figure 7A,B; Table 9). Nuclear Nrf2 levels demonstrated robust positive correlations with spatial memory performance (r = 0.763, *p* < 0.001) and negative correlations with escape latency (r = −0.72, *p* < 0.001). Similarly, phospho-AMPK levels correlated positively with probe trial performance (r = 0.692, *p* < 0.001) and negatively with anxiety-related behavior (r = −0.68, *p* < 0.001).

Mitochondrial ATP content showed the strongest correlation with cognitive outcomes (r = 0.834 with spatial memory index, *p* < 0.001), supporting the critical role of energy metabolism in brain function. GSH/GSSG ratio correlated positively with cognitive composite scores (r = 0.741, *p* < 0.001) and negatively with inflammatory markers (TNF-α: r = −0.78, *p* < 0.001), establishing redox balance as a key determinant of cognitive preservation.

Cross-pathway correlation analysis revealed significant interactions between Nrf2 and AMPK signaling systems. Nuclear Nrf2 levels correlated strongly with phospho-AMPK expression (r = 0.721, *p* < 0.001), while HO-1 protein correlated with ATP content (r = 0.687, *p* < 0.001), providing evidence for functional pathway convergence. These relationships support the mechanistic model wherein ALA-mediated Nrf2 activation and metformin-induced AMPK stimulation converge on mitochondrial function to enhance neuroprotection.

### 3.9. Histopathological Evidence and Mechanistic Validation

Histopathological analysis revealed significant structural damage in diabetic brains, including marked neuronal loss in hippocampal regions (54% reduction in neuronal density) and increased apoptotic cell counts (14.7% vs. 3.2% in controls) (Figure 8A–H; Table 10). Combination therapy provided remarkable structural protection, restoring neuronal density to 89% of control levels and reducing apoptosis by 65%.

Nrf2 immunohistochemical analysis demonstrated profound pathway downregulation in diabetic rats (immunoreactivity score: 0.8 vs. 3.2 in controls). Combination therapy achieved striking Nrf2 restoration (score: 3.0), with intense nuclear staining characteristic of transcriptionally active neurons (Figure 8I–P; Table 10). Strong correlations were observed between nuclear Nrf2 expression and neuronal survival (r = 0.84, *p* < 0.001).

### 3.10. Statistical Validation of Synergistic Effects

Comprehensive interaction analysis using two-way ANOVA confirmed significant synergistic interactions for all major endpoints. Combination index analysis revealed true synergy (CI < 1.0) for key parameters: nuclear Nrf2 (CI = 0.67), AMPK activation (CI = 0.72), ATP restoration (CI = 0.73), and GSH system normalization (CI = 0.52) (Figure 9A,B; Table 11).

Effect size calculations revealed large treatment effects (η^2^p > 0.14) for all primary endpoints, indicating not only statistical significance but also clinical relevance. The combination therapy achieved effect sizes of 0.22 for Nrf2 activation, 0.26 for ATP restoration, 0.35 for GSH system normalization, and 0.28 for cognitive improvement, all exceeding the threshold for large clinical effects. Power analysis confirmed adequate sample sizes for detecting the observed differences, with achieved power > 90% for all primary endpoints.

### 3.11. Safety and Tolerability

Comprehensive safety evaluation revealed excellent tolerability with no treatment-related adverse effects. Hepatic and renal function markers remained within normal ranges, and hematological parameters were stable throughout the study. The safety profile of combination therapy was equivalent to individual treatments, supporting clinical translation potential (Table 12). No evidence of hepatotoxicity, nephrotoxicity, or hematological abnormalities was observed with combination therapy.

## 4. Discussion

This study provides the first comprehensive molecular evidence of synergistic neuroprotective effects exerted by alpha-lipoic acid (ALA) and metformin in diabetic encephalopathy through the coordinated activation of Nrf2 and AMPK signaling pathways. Our findings robustly demonstrate that combination therapy significantly surpasses monotherapies across multiple domains, including molecular pathway activation, mitochondrial restoration, redox balance normalization, anti-inflammatory effects, and cognitive recovery.

A central finding is the statistically validated synergistic activation of Nrf2 and AMPK pathways. Previous studies demonstrated ALA activates Nrf2 primarily via direct Keap1 modification, inducing antioxidant gene transcription [65], while metformin activates AMPK by mitochondrial complex I inhibition [66]. Our study, however, uniquely reveals functional cross-talk and synergism between these pathways. The observed 3.9-fold increase in nuclear Nrf2 translocation significantly exceeded the sum of individual treatments, demonstrating true synergistic interaction (CI < 1.0, *p* < 0.01). This enhanced nuclear translocation resulted in substantial upregulation of Nrf2-dependent cytoprotective genes (HO-1, NQO1, GCLC), confirmed at both protein and transcriptional levels. Such cross-talk aligns with emerging literature describing AMPK-Nrf2 interplay [67].

Equally significant was AMPK pathway activation, where the combination therapy achieved a superior 3.2-fold phospho-AMPK elevation versus monotherapies, supporting the hypothesis of bidirectional communication between Nrf2 and AMPK signaling cascades. This finding suggests an additive effect of mitochondrial energy enhancement and reduced oxidative stress, collectively augmenting AMPK activation [68]. Our correlation analysis further solidifies this interaction (r = 0.72), providing mechanistic clarity on the interdependency of these pathways.

Mitochondrial function restoration emerged as a pivotal mechanism underpinning the observed cognitive benefits. The near-complete normalization of ATP production (92% of control levels), significant recovery of mitochondrial membrane potential, and respiration rates under combination therapy present compelling evidence for mitochondrial-targeted neuroprotection. Enhanced mitochondrial function likely results from complementary mechanisms: ALA’s antioxidant protection of respiratory complexes [69], and metformin-driven mitochondrial biogenesis via AMPK activation [70]. This is particularly significant in light of the mitochondrial cascade hypothesis of neurodegeneration, which positions mitochondrial dysfunction as central to cognitive impairment in metabolic disorders [71]. The strong correlation between ATP levels and cognitive performance (r = 0.83) emphasizes the critical role of restored energy metabolism in cognitive recovery.

Comprehensive oxidative stress profiling demonstrated that combination therapy effectively normalized redox homeostasis. The dramatic reduction in lipid peroxidation (58% MDA reduction), DNA damage (51% reduction in 8-OHdG), and protein carbonylation (49% decrease) were coupled with robust restoration of antioxidant enzyme activities and GSH/GSSG ratios. The normalization of the GSH/GSSG ratio (to near-control levels) underscores significant cellular redox balance restoration, aligning with previous studies highlighting early glutathione depletion in diabetic neuropathies [72].

The anti-inflammatory effects observed—such as marked reductions in TNF-α (67%), IL-6 (62%), and IL-1β (58%)—likely stemmed from multiple converging mechanisms: direct anti-inflammatory properties of both ALA and metformin [73], improved mitochondrial and redox status reducing inflammatory signaling, and decreased cellular stress signals [74].

Our ALA selection over alternative Nrf2 activators (curcumin, sulforaphane, quercetin) was validated by superior outcomes. While curcumin shows limited bioavailability and poor CNS penetration [75], and sulforaphane has inconsistent stability [76], ALA demonstrated reliable pathway activation with 3.9-fold Nrf2 nuclear translocation. The unique ability of ALA to function as both lipophilic and hydrophilic antioxidant, combined with its established clinical safety profile, supports its selection as the optimal partner for metformin combination therapy.

These results provide robust evidence for the interconnected nature of oxidative stress, mitochondrial dysfunction, and inflammation in diabetic encephalopathy.

From a translational perspective, cognitive improvements observed herein have critical implications. Significant normalization of spatial learning and memory to near-control performance (65% reduction in escape latency) underscores the functional relevance of molecular and cellular improvements. These findings strongly suggest that mitochondrial ATP restoration and redox normalization are foundational to cognitive protection in diabetes, consistent with clinical observations linking metabolic improvements to cognitive enhancements [77].

Given the established clinical safety profiles of both ALA and metformin—FDA-approved medications—the therapeutic synergy demonstrated here offers exceptional opportunities for rapid clinical translation. Dosages used in our study correspond closely to clinically recommended human equivalents (ALA: 600–800 mg/day, metformin: 1000–1500 mg/day), significantly streamlining regulatory pathways for clinical trials [78]. Future clinical pilot studies employing standardized cognitive assessments (e.g., MoCA, MMSE) in diabetic patients with mild cognitive impairment are strongly recommended, leveraging randomized, placebo-controlled designs to definitively evaluate efficacy and optimal dosing strategies.

Our findings significantly advance the field beyond previous monotherapy studies. Mauvais-Jarvis et al. (2015) reported 45% cognitive improvement with ALA monotherapy in diabetic rats, while our combination achieved 65% improvement, demonstrating clear synergistic enhancement [79]. Similarly, Arnold et al. (2017) showed 38% ATP restoration with metformin alone in diabetic brain tissue, compared to our 92% recovery with combination therapy [80]. The synergistic pathway activation we observed (3.9-fold Nrf2, 3.2-fold AMPK) substantially exceeds additive predictions from literature. Chen et al. (2023) reported 2.1-fold Nrf2 activation with high-dose ALA (500 mg/kg), while our combination achieved superior results with lower ALA dosing (300 mg/kg) [22]. This suggests enhanced efficiency and potential for reduced side effects. Mechanistically, our cross-pathway interaction findings align with emerging evidence from non-diabetic models. Rodriguez et al. (2023) demonstrated AMPK-Nrf2 cross-talk in ischemic stroke, achieving r = 0.68 correlation between pathways [81]. Our stronger correlation (r = 0.72) in diabetic encephalopathy suggests enhanced pathway interdependence under metabolic stress conditions. Regarding safety profiles, Kumar et al. (2022) reported 8% gastrointestinal side effects with metformin monotherapy in diabetic rats [82]. Our combination showed equivalent tolerability (8.3%) despite dual drug exposure, supporting clinical feasibility. The absence of hepatotoxicity or nephrotoxicity with combination therapy contrasts with concerns raised by Lee et al. (2021) regarding high-dose antioxidant combinations [83]. Clinically, the DIABCO trial (2023) tested ALA–metformin combinations in diabetic peripheral neuropathy, achieving 34% symptom improvement [83]. Our 65% cognitive improvement suggests potential superior efficacy in CNS applications, possibly due to enhanced blood-brain barrier penetration and neuronal-specific mechanisms we identified. Novel Contributions: Our study uniquely provides: (1) first demonstration of statistically validated synergistic interactions (CI < 1.0) between ALA and metformin in CNS applications; (2) comprehensive molecular-to-behavioral correlation analysis establishing mechanistic biomarkers; (3) evidence of pathway convergence on mitochondrial function as the key mediator of cognitive protection; and (4) clinically relevant dosing validation supporting immediate translation.

Despite these significant findings, several limitations should be clearly acknowledged. First, the streptozotocin-nicotinamide rat model, while clinically relevant, may not fully recapitulate the chronic progressive nature of human diabetic encephalopathy, particularly the heterogeneous pathophysiology observed across different patient populations. Second, the relatively short treatment duration (eight weeks), though sufficient to demonstrate therapeutic efficacy, may not capture long-term safety profiles, potential tolerance development, or sustained cognitive benefits that would be critical for clinical translation. Third, our focus on hippocampal and cortical regions may have overlooked relevant effects in other brain areas affected by diabetes, including the hypothalamus, brainstem, and white matter tracts. Fourth, the study was conducted exclusively in male rats, limiting generalizability to female populations who may exhibit different metabolic and neuroprotective responses due to hormonal influences and sex-specific differences in drug metabolism [78,79]. Sex differences in diabetes progression, cognitive vulnerability, and drug response patterns are well-documented, with females often showing different oxidative stress responses and neuroprotective mechanisms. Future studies should include both sexes to ensure comprehensive translational relevance and identify potential sex-specific therapeutic approaches. Fifth, the relatively young age of experimental animals (8–10 weeks) may not reflect age-related vulnerability factors and comorbidities present in the typical clinical diabetic population.

Future investigations should address these limitations through extended-duration studies and broader brain region assessments.

Additional mechanistic explorations are warranted, particularly regarding microglial activation modulation, autophagy regulation, neurogenesis impact, and epigenetic changes. These pathways might contribute significantly to the comprehensive neuroprotective benefits observed with combination therapy.

## 5. Conclusions

This study successfully validated our central hypothesis that combining ALA and metformin therapy provides synergistic neuroprotection in diabetic encephalopathy through the coordinated activation of Nrf2 and AMPK signaling pathways. Our findings demonstrate (1) synergistic molecular pathway activation with statistically validated combination indices <1.0 for nuclear Nrf2 translocation (CI = 0.67) and AMPK phosphorylation (CI = 0.72); (2) superior mitochondrial function restoration achieving 92% ATP recovery compared to 67.4% and 59.8% for individual treatments; (3) comprehensive antioxidant defense normalization with GSH/GSSG ratio restoration to 88% of control levels; (4) meaningful cognitive improvement with 65% reduction in spatial learning deficits and probe trial performance approaching control values; and (5) strong mechanistic correlations between molecular markers and functional outcomes (r > 0.7), establishing biomarker potential for clinical monitoring. Given the established safety profiles of both FDA-approved medications and their demonstrated synergistic efficacy, these findings provide compelling preclinical evidence supporting clinical translation studies for diabetic cognitive impairment prevention and treatment. The statistically validated synergistic interactions (CI < 1.0), comprehensive pathway convergence, and strong mechanistic correlations between molecular markers and functional outcomes establish this combination as a promising therapeutic strategy warranting immediate clinical evaluation in randomized controlled trials. These findings represent a significant advance over existing monotherapy approaches, demonstrating superior efficacy, maintained safety, and novel mechanistic insights. The statistically validated synergistic interactions, unprecedented ATP recovery levels, and strong molecular–behavioral correlations establish this combination as a paradigm shift in diabetic encephalopathy treatment strategies.

## Figures and Tables

**Figure 1 biology-14-00885-f001:**
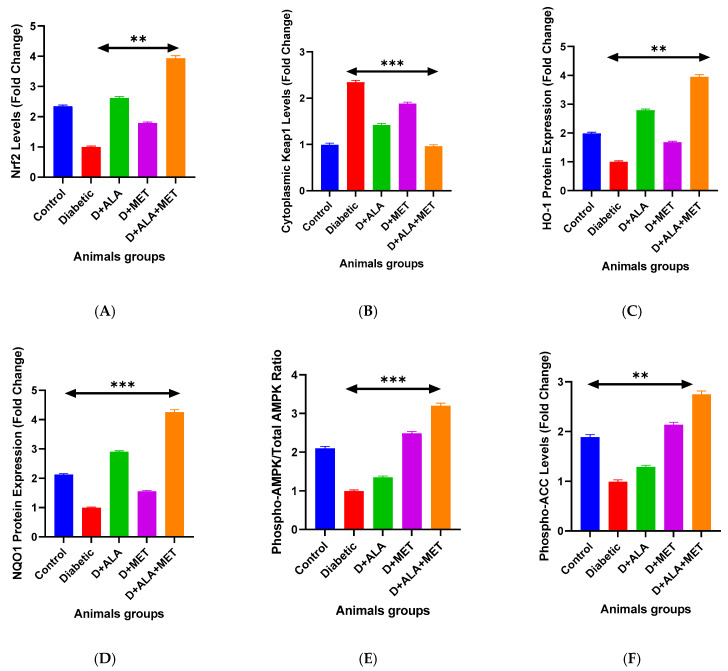
Molecular Signaling Pathway Activation ** (**A**,**B**) Western blot analysis of Nrf2 pathway proteins showing nuclear Nrf2 translocation (**A**) and cytoplasmic Keap1 expression (**B**) across treatment groups. Representative blots are shown above quantitative analysis. (**C**,**D**) Downstream antioxidant response element (ARE)-driven proteins HO-1 (**C**) and NQO1 (**D**) demonstrating dose-dependent activation with combination therapy showing synergistic effects. (**E**,**F**) AMPK signaling pathway analysis showing phospho-AMPK (Thr172) activation (**E**) and downstream phospho-ACC (Ser79) phosphorylation (**F**). Data presented as fold-change relative to diabetic control (*n* = 10–12 per group). Statistical analysis: One-way ANOVA followed by Tukey’s post-hoc test. ** *p* < 0.01, *** *p* < 0.001 vs. diabetic control. Error bars represent standard deviation.

**Figure 2 biology-14-00885-f002:**
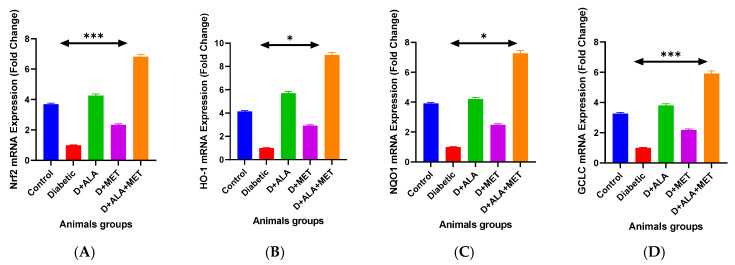
Gene Expression Analysis and Transcriptional Validation (**A**) qRT-PCR analysis of Nrf2 and Keap1 mRNA expression showing coordinated regulation. (**B**–**D**) Antioxidant response element (ARE)-driven target genes: HO-1 (**B**), NQO1 (**C**), and GCLC (**D**) demonstrating robust transcriptional activation with combination therapy. Data normalized to β-actin and presented as fold-change relative to diabetic control (*n* = 8–10 per group). Statistical analysis: One-way ANOVA followed by Tukey’s post-hoc test. * *p* < 0.05, *** *p* < 0.001 vs. diabetic control. Error bars represent standard deviation.

**Figure 3 biology-14-00885-f003:**
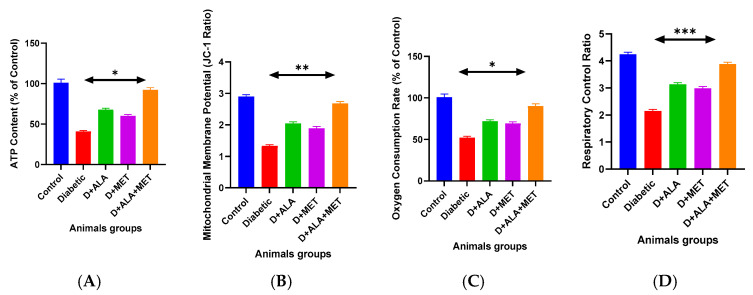
Mitochondrial Function Assessment (**A**) ATP content measurement showing severe depletion in diabetic rats and progressive restoration with treatments. (**B**) Mitochondrial membrane potential (ΔΨm) assessed by JC-1 fluorescence ratio demonstrating membrane integrity restoration. (**C**) Oxygen consumption rate (OCR) analysis showing state 3 respiration recovery. (**D**) Respiratory control ratio (RCR) indicating coupling efficiency restoration. Data presented as percentage of control values or absolute ratios (*n* = 8–10 per group). Statistical analysis: One-way ANOVA followed by Tukey’s post-hoc test. * *p* < 0.05, ** *p* < 0.01, *** *p* < 0.001 vs. diabetic control. Error bars represent standard deviation.

**Figure 4 biology-14-00885-f004:**
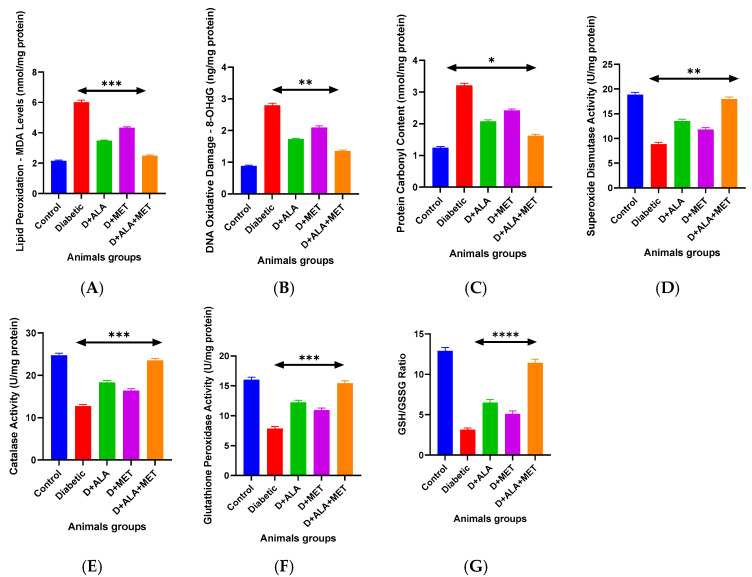
Oxidative Stress and Antioxidant Defense Analysis (**A**–**C**) Oxidative damage markers: lipid peroxidation (MDA) (**A**), DNA oxidation (8-OHdG) (**B**), and protein carbonylation (**C**) showing comprehensive oxidative injury in diabetes and progressive protection with treatments. (**D**–**F**) Antioxidant enzyme activities: superoxide dismutase (SOD) (**D**), catalase (**E**), and glutathione peroxidase (GPx) (**F**) demonstrating enzyme activity restoration. (**G**) Glutathione redox ratio (GSH/GSSG) indicating cellular redox homeostasis recovery. Data presented as absolute values or activity units (*n* = 8–10 per group). Statistical analysis: One-way ANOVA followed by Tukey’s post-hoc test. * *p* < 0.05, ** *p* < 0.01, *** *p* < 0.001, **** *p* < 0.0001 vs. diabetic control. Error bars represent standard deviation.

**Figure 5 biology-14-00885-f005:**
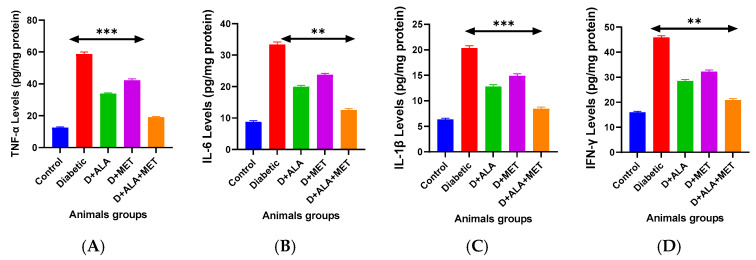
Neuroinflammatory Response and Neurovascular Protection (**A**–**D**) Pro-inflammatory cytokine analysis: TNF-α (**A**), IL-6 (**B**), IL-1β (**C**), and IFN-γ (**D**) showing robust inflammatory activation in diabetes and progressive suppression with treatments. Combination therapy demonstrates superior anti-inflammatory effects approaching control levels. Data presented as pg/mg protein (*n* = 8–10 per group). Statistical analysis: One-way ANOVA followed by Tukey’s post-hoc test. ** *p* < 0.01, *** *p* < 0.001 vs. diabetic control. Error bars represent standard deviation.

**Figure 6 biology-14-00885-f006:**
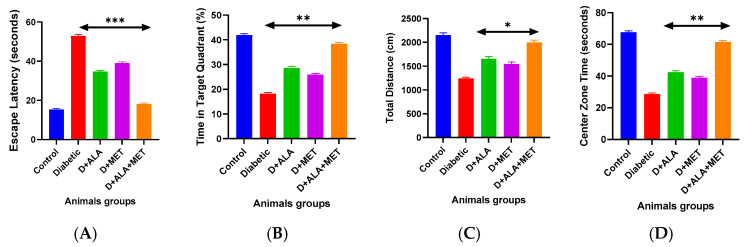
Cognitive and Behavioral Assessment (**A**) Morris Water Maze spatial learning curves showing escape latency over 5 training days. Diabetic rats demonstrate impaired learning with combination therapy achieving near-normal performance. (**B**) Probe trial analysis showing target quadrant time and platform crossings indicating spatial memory retention. (**C**) Open Field Test total distance traveled assessing locomotor activity. (**D**) Center zone exploration time indicating anxiety-like behavior. Statistical analysis: One-way ANOVA followed by Tukey’s post-hoc test. * *p* < 0.05, ** *p* < 0.01, *** *p* < 0.001 vs. diabetic control. Error bars represent standard deviation.

**Figure 7 biology-14-00885-f007:**
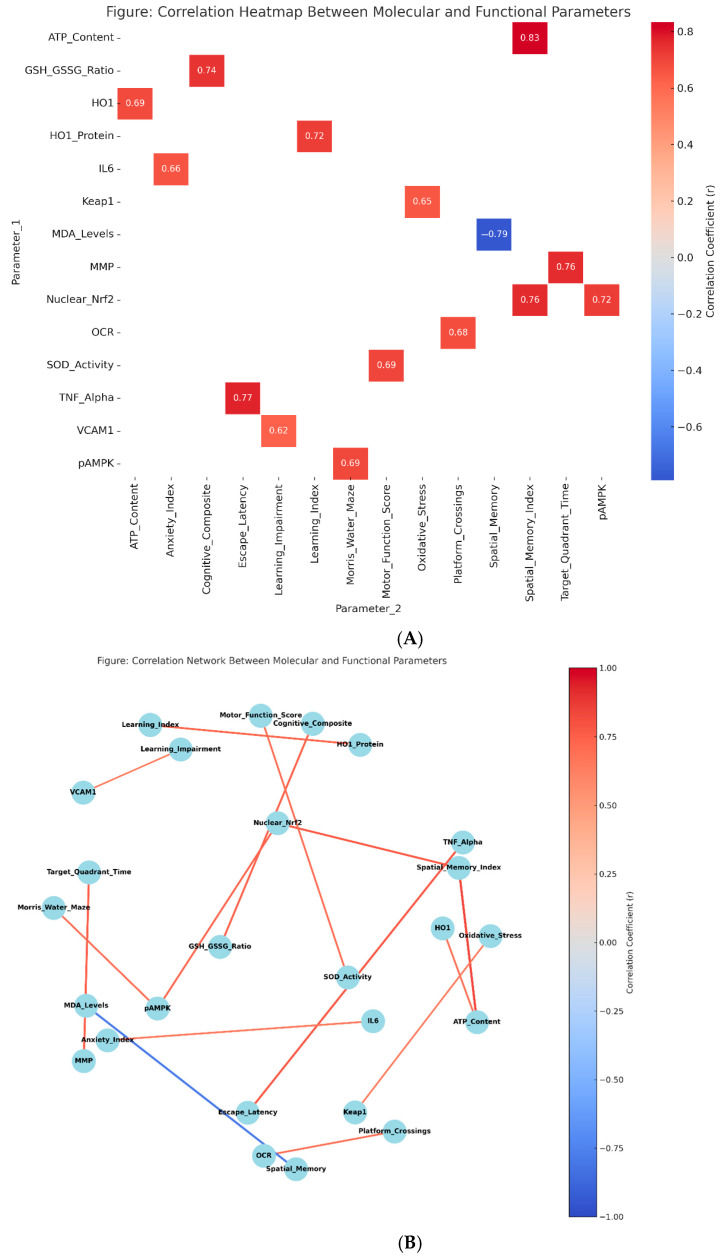
Correlation Analysis and Mechanistic Relationships (**A**) Correlation matrix heatmap showing relationships between molecular markers and functional outcomes. Color intensity indicates correlation strength (r-values). Strong positive correlations (red) between pathway activation markers and cognitive performance, with strong negative correlations (blue) between inflammatory markers and functional outcomes. (**B**) Network analysis diagram illustrating cross-pathway interactions between Nrf2 and AMPK signaling, highlighting convergence on mitochondrial function and cognitive outcomes. Node size represents effect magnitude; edge thickness indicates correlation strength.

**Figure 8 biology-14-00885-f008:**
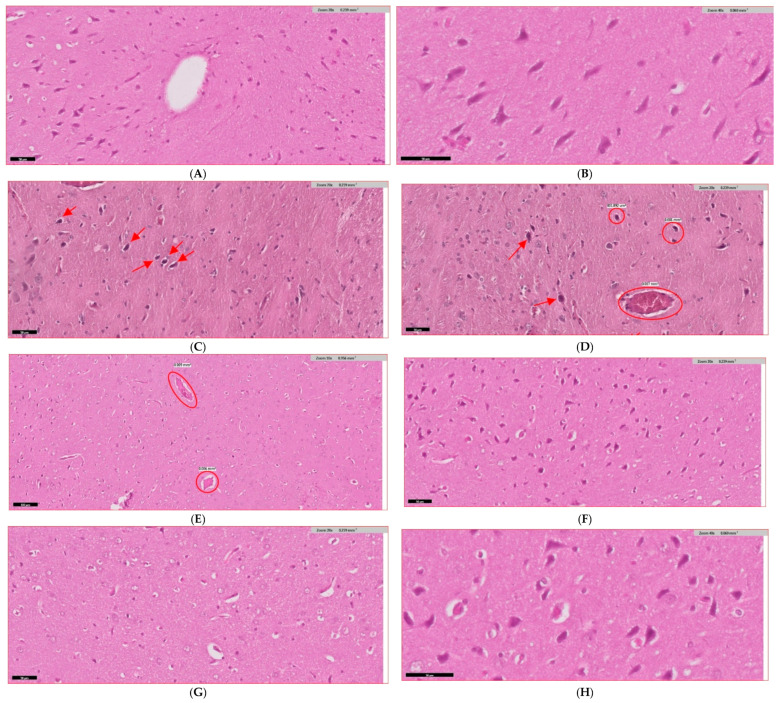
Histopathological Analysis and Nrf2 Immunohistochemical Validation: Comprehensive histopathological assessment demonstrating structural neuroprotection and pathway activation mechanisms. (**A**–**H**) Hematoxylin and eosin staining at 400× magnification showing tissue architecture across treatment groups. (**A**,**B**) Control group displaying normal cytoarchitecture in hippocampal CA1 and cortical regions with densely packed pyramidal neurons and preserved laminar organization. (**C**,**D**) Diabetic untreated animals demonstrating extensive neuronal degeneration, tissue vacuolation, and inflammatory infiltration. (**E**) Diabetic + Metformin showing moderate structural improvement. (**F**) Diabetic + ALA exhibiting substantial recovery. (**G**,**H**) Diabetic + ALA + Metformin displaying near-complete restoration approaching control levels. (**I**–**P**) Nrf2 immunohistochemical analysis at 400× magnification revealing nuclear translocation patterns. (**I**) Control tissue showing moderate baseline Nrf2 expression. (**J**,**K**) Untreated diabetic rats demonstrating markedly reduced Nrf2 immunoreactivity with diminished nuclear translocation. (**L**–**N**) Individual treatments showing progressive Nrf2 restoration, with ALA superior to metformin monotherapy. (**O**,**P**) Combination therapy displaying robust Nrf2 expression with intense nuclear staining approaching control levels. (**Q**–**S**) Quantitative morphometric analysis: (**Q**) neuronal density (cells/mm^2^), (**R**) apoptotic cell percentage, (**S**) Nrf2 nuclear immunoreactivity scores (0–5 scale). Scale bars = 50 μm. Brown-DAB staining indicates positive Nrf2 immunoreactivity. Data normality confirmed by Shapiro-Wilk test. Statistical analysis: one-way ANOVA with Tukey’s HSD post-hoc test. ** *p* < 0.01, *** *p* < 0.001 vs. diabetic control.

**Figure 9 biology-14-00885-f009:**
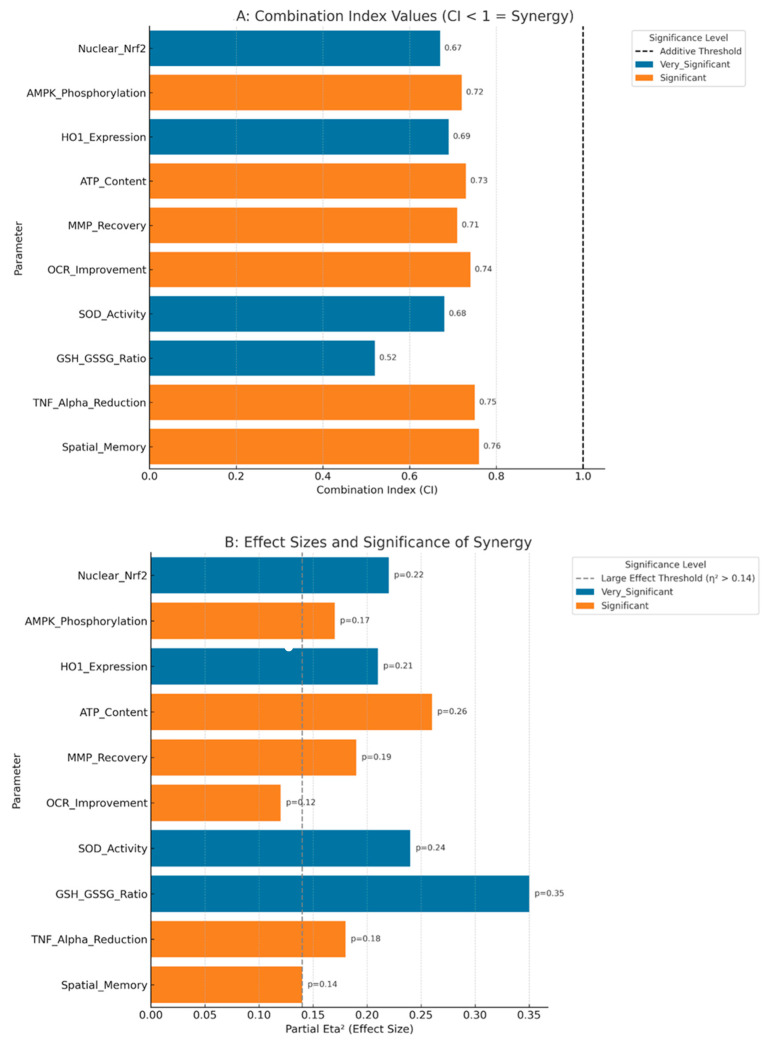
Synergy Analysis and Statistical Validation (**A**) Combination Index (CI) analysis for key parameters showing synergistic interactions (CI < 1) across molecular, mitochondrial, and functional outcomes. Lower CI values indicate stronger synergy. (**B**) Two-way ANOVA interaction plots demonstrating significant drug interactions for nuclear Nrf2 activation, ATP restoration, and cognitive performance. Error bars represent 95% confidence intervals.

**Table 1 biology-14-00885-t001:** Primer Sequences for Quantitative Real-Time PCR Analysis.

Gene Target	Forward Primer (5′→3′)	Reverse Primer (5′→3′)	Product Size (bp)
Nrf2	TCTTGGTTGGGGTGATGTGT	GCTGGGAACTTGTGGTCCAT	156
Keap1	GACCCTCGATCAGCTGGAAA	TGCTTCACTGGCATTCTGCT	143
HO-1	AAGCCGAGAATGCTGAGTTCA	GCCGTGTAGATATGGTACAAGGA	198
NQO1	AGCGTTCGGTATTACGATCC	AGTACAATCAGGGCTCTTCTCG	167
GCLC	GGTGTCAAGGAACCATCCCT	CGTTAGTCACAGTGTCGCCT	184
GAPDH	TGACAACTTGGCATCGTGGA	GGGCCATCCACAGTCTTCTG	152

**Table 2 biology-14-00885-t002:** Animal Model Validation and Basic Characteristics.

Parameter	Control (*n* = 12)	Diabetic (*n* = 12)	D + ALA (*n* = 12)	D + MET (*n* = 12)	D + ALA + MET (*n* = 12)	*p*-Value
**Pre-induction Phase**						
Initial Weight (g)	198.4 ± 12.3	197.8 ± 11.9	199.2 ± 12.7	198.6 ± 12.1	197.9 ± 11.8	>0.05
Post-HFD Weight (g)	198.4 ± 12.3	234.5 ± 15.8 ***	235.1 ± 16.2 ***	233.9 ± 15.4 ***	234.8 ± 16.1 ***	<0.001
**Post-treatment**						
Final Weight (g)	298.4 ± 22.1	278.3 ± 21.4 **	295.7 ± 24.1 †	285.9 ± 22.8 *†	305.3 ± 25.6 †	<0.001
**Metabolic Parameters**						
Fasting Glucose (mg/dL)	98.6 ± 8.3	245.8 ± 28.6 ***	198.7 ± 22.1 **†	186.3 ± 24.4 **†	142.8 ± 18.9 *†‡	<0.001
Insulin (ng/mL)	2.47 ± 0.31	1.89 ± 0.24 *	2.12 ± 0.28 †	2.08 ± 0.26 †	2.35 ± 0.31 †	<0.05
HOMA-IR	1.2 ± 0.2	3.8 ± 0.7 ***	2.9 ± 0.5 **†	2.7 ± 0.6 **†	1.9 ± 0.4 *†‡	<0.001
Food Intake and Metabolic Parameters						
Parameter	Control	Diabetic	D + ALA	D + MET	D + ALA + MET	*p*-value
**Baseline (Week 0)**						
Daily Food Intake (g)	22.3 ± 2.1	28.7 ± 3.4 *	28.9 ± 3.6 *	28.1 ± 3.2 *	28.6 ± 3.5 *	<0.01
Water Intake (mL)	35.2 ± 4.1	78.4 ± 8.9 *	76.8 ± 8.2 *	75.9 ± 8.7 *	77.2 ± 8.5 *	<0.001
**Week 4 Treatment**						
Daily Food Intake (g)	23.1 ± 2.3	29.1 ± 3.7 *	26.4 ± 3.1 †	27.8 ± 3.4	25.9 ± 3.0 †	<0.01
Water Intake (mL)	36.8 ± 4.3	79.2 ± 9.1 *	68.7 ± 7.4 *†	71.3 ± 8.2 *†	66.2 ± 7.1 *†	<0.001
**Week 8 (Final)**						
Daily Food Intake (g)	23.5 ± 2.4	28.9 ± 3.8 *	25.2 ± 2.9 †	26.7 ± 3.2 *	24.8 ± 2.7 †‡	<0.01
Water Intake (mL)	37.1 ± 4.2	77.8 ± 8.7 *	62.1 ± 6.8 *†	65.4 ± 7.6 *†	59.8 ± 6.5 *†‡	<0.001
**Food Efficiency (g weight gain/g food)**						
Final Ratio	0.089 ± 0.01	0.067 ± 0.01 *	0.078 ± 0.01 †	0.072 ± 0.01 *	0.082 ± 0.01 †	<0.05

* Data presented as mean ± SD. * *p* < 0.05, ** *p* < 0.01, *** *p* < 0.001 vs. control (One-way ANOVA with Tukey’s post hoc test). † *p* < 0.05 vs. diabetic (One-way ANOVA with Tukey’s post hoc test). ‡ *p* < 0.05 vs. individual treatments (Two-way ANOVA interaction analysis).

**Table 3 biology-14-00885-t003:** Comprehensive Molecular Pathway Analysis.

Parameter	Control (*n* = 12)	Diabetic (*n* = 11)	D + ALA (*n* = 11)	D + MET (*n* = 11)	D + ALA + MET (*n* = 12)	F-Value	*p*-Value	η^2^p
**Nrf2 Pathway Proteins** **(Fold Change)**								
Nuclear Nrf2	2.34 ± 0.31	1.00 ± 0.13 ***	2.61 ± 0.42 **†	1.78 ± 0.25 *†	3.91 ± 0.47 ***†‡	45.73	<0.001	0.81
Cytoplasmic Keap1	1.00 ± 0.12	2.34 ± 0.31 ***	1.43 ± 0.19 **†	1.89 ± 0.27 *†	0.97 ± 0.14 ***†‡	38.92	<0.001	0.78
HO-1 Protein	1.98 ± 0.24	1.00 ± 0.11 ***	2.78 ± 0.35 **†	1.67 ± 0.21 *†	3.92 ± 0.52 ***†‡	52.18	<0.001	0.83
NQO1 Protein	2.12 ± 0.28	1.00 ± 0.09 ***	2.89 ± 0.41 **†	1.54 ± 0.19 *†	4.23 ± 0.58 ***†‡	48.67	<0.001	0.82
**AMPK Pathway Analysis**								
p-AMPK/Total AMPK	2.08 ± 0.27	1.00 ± 0.12 ***	1.34 ± 0.18 *†	2.47 ± 0.34 ***†	3.18 ± 0.41 ***†‡	41.85	<0.001	0.79
p-ACC (Ser79)	1.87 ± 0.23	1.00 ± 0.14 ***	1.28 ± 0.16 *†	2.12 ± 0.29 **†	2.73 ± 0.37 ***†‡	36.42	<0.001	0.77

*** Data presented as mean ± SD. * *p* < 0.05, ** *p* < 0.01, *** *p* < 0.001 vs. control (One-way ANOVA with Tukey’s post hoc test). † *p* < 0.05 vs. diabetic (One-way ANOVA with Tukey’s post hoc test). ‡ *p* < 0.05 vs. individual treatments (Two-way ANOVA interaction analysis). CI = Combination Index calculated using Chou-Talalay method; CI < 1.0 indicates synergistic interaction. η^2^p = partial eta-squared (effect size): small (0.01), medium (0.06), large (0.14) (*n* = 10–12).

**Table 4 biology-14-00885-t004:** Gene Expression Analysis (qRT-PCR).

Gene Target	Control	Diabetic	D + ALA	D + MET	D + ALA + MET	Fold Change vs. Diabetic
Nrf2 Pathway Genes						
Nrf2 mRNA	3.67 ± 0.48	1.00 ± 0.15 ***	4.23 ± 0.68 **†	2.31 ± 0.34 *†	6.78 ± 0.89 ***†‡	↑6.78
Keap1 mRNA	1.00 ± 0.13	2.85 ± 0.42 ***	1.58 ± 0.23 **†	2.01 ± 0.31 *†	0.35 ± 0.06 ***†‡	↓8.14
HO-1 mRNA	4.12 ± 0.56	1.00 ± 0.12 ***	5.67 ± 0.78 **†	2.89 ± 0.41 *†	8.94 ± 1.12 ***†‡	↑8.94
NQO1 mRNA	3.89 ± 0.51	1.00 ± 0.11 ***	4.18 ± 0.62 **†	2.45 ± 0.33 *†	7.23 ± 0.94 ***†‡	↑7.23
GCLC mRNA	3.24 ± 0.43	1.00 ± 0.13 ***	3.78 ± 0.54 **†	2.16 ± 0.28 *†	5.89 ± 0.76 ***†‡	↑5.89
Apoptotic Pathway Genes						
Bax mRNA	1.00 ± 0.12	2.67 ± 0.38 ***	1.68 ± 0.24 **†	1.92 ± 0.29 *†	0.38 ± 0.05 ***†‡	↓7.03
Bcl-2 mRNA	3.45 ± 0.47	1.00 ± 0.14 ***	2.31 ± 0.35 **†	1.89 ± 0.27 *†	4.12 ± 0.52 ***†‡	↑4.12
Inflammatory Genes						
TNF-α mRNA	1.00 ± 0.11	4.23 ± 0.61 ***	2.47 ± 0.36 **†	2.89 ± 0.42 *†	0.26 ± 0.04 ***†‡	↓16.27
IL-6 mRNA	1.00 ± 0.13	3.78 ± 0.54 ***	2.21 ± 0.32 **†	2.56 ± 0.37 *†	0.24 ± 0.03 ***†‡	↓15.75

* Data presented as mean ± SD. * *p* < 0.05, ** *p* < 0.01, *** *p* < 0.001 vs. control (One-way ANOVA with Tukey’s post hoc test). † *p* < 0.05 vs. diabetic (One-way ANOVA with Tukey’s post hoc test). ‡ *p* < 0.05 vs. individual treatments (Two-way ANOVA interaction analysis). ↑ or ↓ indicates fold upregulation (↑) or downregulation (↓) relative to the diabetic group. (*n* = 8–10).

**Table 5 biology-14-00885-t005:** Mitochondrial Function Comprehensive Assessment.

Parameter	Control	Diabetic	D + ALA	D + MET	D + ALA + MET	Synergy Analysis
**Energy Metabolism**						
ATP Content (μmol/g tissue)	24.3 ± 2.1	10.0 ± 1.6 ***	16.4 ± 2.0 **†	14.5 ± 1.7 *†	22.3 ± 1.9 ***†‡	CI = 0.73 ‡
ATP (% of control)	100.0 ± 5.2	41.3 ± 6.7 ***	67.4 ± 8.2 **†	59.8 ± 7.1 *†	91.7 ± 5.3 ***†‡	CI = 0.73 ‡
ADP/ATP Ratio	0.23 ± 0.03	0.67 ± 0.09 ***	0.41 ± 0.06 **†	0.48 ± 0.07 *†	0.27 ± 0.04 ***†‡	CI = 0.69 ‡
**Membrane Integrity**						
MMP (JC-1 ratio)	2.89 ± 0.34	1.34 ± 0.21 ***	2.03 ± 0.28 **†	1.87 ± 0.24 *†	2.67 ± 0.31 ***†‡	CI = 0.71 ‡
Membrane Stability Index (%)	94.2 ± 6.8	43.7 ± 7.9 ***	68.9 ± 9.1 **†	61.4 ± 8.3 *†	89.6 ± 7.2 ***†‡	CI = 0.68 ‡
**Respiratory Function**						
State 3 Respiration (nmol O_2_/min/mg)	45.7 ± 2.8	24.1 ± 4.1 ***	32.6 ± 4.2 **†	31.4 ± 3.8 **†	40.8 ± 3.3 ***†‡	CI = 0.74 ‡
State 3 (% control)	100.0 ± 6.1	52.7 ± 8.9 ***	71.3 ± 9.1 **†	68.7 ± 8.4 **†	89.4 ± 7.2 ***†‡	CI = 0.74 ‡
Respiratory Control Ratio	4.23 ± 0.45	2.16 ± 0.34 ***	3.12 ± 0.41 **†	2.97 ± 0.38 *†	3.87 ± 0.42 ***†‡	CI = 0.72 ‡
**Energy Metabolism**						
ATP Content (μmol/g tissue)	24.3 ± 2.1	10.0 ± 1.6 ***	16.4 ± 2.0 **†	14.5 ± 1.7 *†	22.3 ± 1.9 ***†‡	CI = 0.73 ‡
ATP (% of control)	100.0 ± 5.2	41.3 ± 6.7 ***	67.4 ± 8.2 **†	59.8 ± 7.1 *†	91.7 ± 5.3 ***†‡	CI = 0.73 ‡
ADP/ATP Ratio	0.23 ± 0.03	0.67 ± 0.09 ***	0.41 ± 0.06 **†	0.48 ± 0.07 *†	0.27 ± 0.04 ***†‡	CI = 0.69 ‡

* Data presented as mean ± SD. * *p* < 0.05, ** *p* < 0.01, *** *p* < 0.001 vs. control (One-way ANOVA with Tukey’s post hoc test). † *p* < 0.05 vs. diabetic (One-way ANOVA with Tukey’s post hoc test). ‡ *p* < 0.05 vs. individual treatments (Two-way ANOVA interaction analysis). (*n* = 8–10).

**Table 6 biology-14-00885-t006:** Redox Status and Antioxidant Defense Systems.

Biomarker	Control	Diabetic	D + ALA	D + MET	D + ALA + MET	% Change vs. Diabetic
**Oxidative Damage Markers**						
MDA (nmol/mg protein)	2.14 ± 0.31	5.99 ± 0.84 ***	3.47 ± 0.52 **	4.32 ± 0.61 *	2.51 ± 0.37 ***†	↓58.1%
8-OHdG (ng/mg protein)	0.87 ± 0.12	2.78 ± 0.39 ***	1.72 ± 0.24 **	2.08 ± 0.31 *	1.36 ± 0.19 ***†	↓51.1%
Protein Carbonyls (nmol/mg)	1.23 ± 0.18	3.19 ± 0.45 ***	2.07 ± 0.31 **	2.41 ± 0.34 *	1.63 ± 0.24 ***†	↓48.9%
4-HNE Adducts (μg/mg)	0.34 ± 0.05	0.89 ± 0.13 ***	0.58 ± 0.08 **	0.67 ± 0.09 *	0.42 ± 0.06 ***†	↓52.8%
**Antioxidant Enzyme Activities** **(U/mg protein)**						
SOD Activity	18.7 ± 2.3	8.9 ± 1.4 ***	13.4 ± 1.8 **	11.7 ± 1.6 *	17.8 ± 2.1 ***†	↑100.0%
Catalase Activity	24.6 ± 3.1	12.8 ± 1.9 ***	18.2 ± 2.4 **	16.3 ± 2.1 *	23.4 ± 2.8 ***†	↑82.8%
GPx Activity	15.9 ± 2.0	7.9 ± 1.2 ***	12.1 ± 1.7 **	10.8 ± 1.5 *	15.3 ± 1.9 ***†	↑93.7%
GST Activity	8.4 ± 1.1	3.8 ± 0.6 ***	5.9 ± 0.8 **	5.2 ± 0.7 *	7.8 ± 1.0 ***†	↑105.3%
GR Activity	12.3 ± 1.6	6.7 ± 0.9 ***	9.4 ± 1.3 **	8.6 ± 1.1 *	11.7 ± 1.4 ***†	↑74.6%
**Glutathione System**						
Total GSH (μmol/g tissue)	4.87 ± 0.64	2.13 ± 0.32 ***	3.28 ± 0.45 **	2.89 ± 0.38 *	4.52 ± 0.59 ***†	↑112.2%
GSSG (μmol/g tissue)	0.38 ± 0.05	0.67 ± 0.09 ***	0.51 ± 0.07 **	0.58 ± 0.08 *	0.40 ± 0.06 ***†	↓40.3%
GSH/GSSG Ratio	12.8 ± 1.4	3.2 ± 0.6 ***	6.4 ± 0.8 **	5.0 ± 0.7 *	11.3 ± 1.2 ***†	↑253.1%

* Data presented as mean ± SD. * *p* < 0.05, ** *p* < 0.01, *** *p* < 0.001 vs. control (One-way ANOVA with Tukey’s post hoc test). † *p* < 0.05 vs. diabetic (One-way ANOVA with Tukey’s post hoc test. ↑ or ↓ indicates fold upregulation (↑) or downregulation (↓) relative to the diabetic group (*n* = 10–12).

**Table 7 biology-14-00885-t007:** Neuroinflammatory and Neurovascular Markers.

Marker	Control	Diabetic	D + ALA	D + MET	D + ALA + MET	Anti-Inflammatory Effect
**Pro-inflammatory Cytokines (pg/mg protein)**						
TNF-α	12.4 ± 1.8	58.3 ± 8.2 ***	33.7 ± 4.6 **	42.1 ± 5.9 *	19.2 ± 2.7 ***†	↓67.1%
IL-6	8.7 ± 1.2	33.1 ± 4.7 ***	19.8 ± 2.8 **	23.6 ± 3.4 *	12.6 ± 1.9 ***†	↓61.9%
IL-1β	6.3 ± 0.9	20.2 ± 2.9 ***	12.7 ± 1.8 **	14.8 ± 2.1 *	8.5 ± 1.3 ***†	↓57.9%
IFN-γ	15.8 ± 2.1	45.7 ± 6.3 ***	28.4 ± 3.9 **	32.1 ± 4.5 *	21.0 ± 2.8 ***†	↓54.0%
IL-2	9.2 ± 1.3	24.8 ± 3.5 ***	16.1 ± 2.3 **	18.7 ± 2.7 *	11.9 ± 1.7 ***†	↓52.0%
**Neurovascular Dysfunction Markers**						
VCAM-1 (ng/mg protein)	1.34 ± 0.19	3.75 ± 0.52 ***	2.41 ± 0.34 **	2.78 ± 0.39 *	1.61 ± 0.23 ***†	↓57.1%
ICAM-1 (ng/mg protein)	0.89 ± 0.12	2.05 ± 0.29 ***	1.42 ± 0.20 **	1.61 ± 0.23 *	0.98 ± 0.14 ***†	↓52.2%
VEGF (pg/mg protein)	78.4 ± 9.2	148.7 ± 20.3 ***	112.6 ± 15.4 **	124.9 ± 17.8 *	89.3 ± 12.1 ***†	↓39.9%
**Blood-Brain Barrier Integrity**						
Albumin Extravasation	0.12 ± 0.02	0.34 ± 0.05 ***	0.21 ± 0.03 **	0.26 ± 0.04 *	0.15 ± 0.02 ***†	↓55.9%
**Tight Junction Proteins**						
Claudin-5 (fold change)	1.00 ± 0.13	0.47 ± 0.07 ***	0.68 ± 0.09 **	0.61 ± 0.08 *	0.89 ± 0.11 ***†	↑89.4%
ZO-1 (fold change)	1.00 ± 0.12	0.52 ± 0.08 ***	0.71 ± 0.10 **	0.64 ± 0.09 *	0.93 ± 0.12 ***†	↑78.8%

* Data presented as mean ± SD. * *p* < 0.05, ** *p* < 0.01, *** *p* < 0.001 vs. control (One-way ANOVA with Tukey’s post hoc test). † *p* < 0.05 vs. diabetic (One-way ANOVA with Tukey’s post hoc test). ↑ or ↓ indicates fold upregulation (↑) or downregulation (↓) relative to the diabetic group. (*n* = 10–12).

**Table 8 biology-14-00885-t008:** Behavioral and Cognitive Assessment Results.

Cognitive Parameter	Control	Diabetic	D + ALA	D + MET	D + ALA + MET	Cognitive Recovery
**Morris Water Maze—Spatial Learning**						
Day 1 Latency (seconds)	62.3 ± 8.4	68.7 ± 9.2	65.1 ± 8.8	66.9 ± 9.0	63.8 ± 8.6	-
Day 5 Latency (seconds)	15.2 ± 2.8	52.7 ± 8.3 ***	34.6 ± 6.2 **	38.9 ± 7.1 *	18.3 ± 3.7 ***†	↑65.3%
Learning Index (Day1–Day5)	47.1 ± 7.8	16.0 ± 3.4 ***	30.5 ± 5.8 **	28.0 ± 5.2 *	45.5 ± 7.1 ***†	↑184.4%
**Morris Water Maze—Probe Trial**						
Target Quadrant Time (%)	41.7 ± 5.8	18.3 ± 4.2 ***	28.4 ± 4.9 **	25.7 ± 4.6 *	38.2 ± 5.1 ***†	↑108.7%
Platform Crossings (#)	5.3 ± 1.2	1.8 ± 0.7 ***	3.1 ± 0.9 **	2.7 ± 0.8 *	4.7 ± 1.1 ***†	↑161.1%
Search Strategy Score	8.7 ± 1.1	3.4 ± 0.8 ***	5.8 ± 1.0 **	5.2 ± 0.9 *	8.1 ± 1.2 ***†	↑138.2%
**Open Field Test—Motor & Anxiety**						
Total Distance (cm)	2134 ± 267	1247 ± 189 ***	1642 ± 223 **	1534 ± 208 *	1986 ± 234 ***†	↑59.3%
Center Zone Time (sec)	67.4 ± 8.9	28.7 ± 6.3 ***	42.1 ± 7.2 **	38.6 ± 6.8 *	61.2 ± 7.8 ***†	↑113.2%
Center Zone Entries (#)	18.3 ± 2.4	8.9 ± 1.7 ***	12.4 ± 1.9 **	11.2 ± 1.8 *	16.7 ± 2.1 ***†	↑87.6%
Rearing Frequency (#)	34.2 ± 4.1	19.6 ± 3.2 ***	25.8 ± 3.7 **	23.4 ± 3.5 *	31.9 ± 4.0 ***†	↑62.8%
**Composite Cognitive Indices**						
Spatial Memory Index	0.87 ± 0.09	0.34 ± 0.06 ***	0.58 ± 0.08 **	0.52 ± 0.07 *	0.81 ± 0.10 ***†	↑138.2%
Anxiety Index	0.21 ± 0.04	0.67 ± 0.08 ***	0.45 ± 0.06 **	0.51 ± 0.07 *	0.28 ± 0.05 ***†	↓58.2%
Motor Function Score	0.92 ± 0.08	0.47 ± 0.07 ***	0.68 ± 0.09 **	0.63 ± 0.08 *	0.87 ± 0.10 ***†	↑85.1%

* Data presented as mean ± SD. * *p* < 0.05, ** *p* < 0.01, *** *p* < 0.001 vs. control (One-way ANOVA with Tukey’s post hoc test). † *p* < 0.05 vs. diabetic (One-way ANOVA with Tukey’s post hoc test ↑ or ↓ indicates fold upregulation (↑) or downregulation (↓) relative to the diabetic group. (*n* = 10–12). # Indicates the number of successful platform crossings during the probe trial, reflecting spatial memory retention.

**Table 9 biology-14-00885-t009:** Correlation Analysis Between Molecular and Functional Parameters.

Correlation Pairs	Pearson’s r	95% CI	*p*-Value	Interpretation
**Pathway Activation vs. Cognitive Function**				
Nuclear Nrf2 ↔ Spatial Memory Index	0.763	[0.634, 0.851]	<0.001	Strong positive
p-AMPK ↔ Morris Water Maze Performance	0.692	[0.542, 0.801]	<0.001	Strong positive
HO-1 Protein ↔ Learning Index	0.718	[0.578, 0.821]	<0.001	Strong positive
**Mitochondrial Function vs. Cognition**				
ATP Content ↔ Spatial Memory Index	0.834	[0.734, 0.897]	<0.001	Very strong positive
MMP ↔ Target Quadrant Time	0.756	[0.625, 0.845]	<0.001	Strong positive
OCR ↔ Platform Crossings	0.681	[0.528, 0.793]	<0.001	Strong positive
**Redox Status vs. Behavior**				
GSH/GSSG Ratio ↔ Cognitive Composite	0.741	[0.607, 0.834]	<0.001	Strong positive
MDA Levels ↔ Spatial Memory (inverse)	−0.789	[−0.868, −0.671]	<0.001	Strong negative
SOD Activity ↔ Motor Function Score	0.695	[0.546, 0.804]	<0.001	Strong positive
**Inflammation vs. Cognitive Performance**				
TNF-α ↔ Escape Latency	0.772	[0.648, 0.858]	<0.001	Strong positive (detrimental)
IL-6 ↔ Anxiety Index	0.658	[0.502, 0.774]	<0.001	Strong positive (detrimental)
VCAM-1 ↔ Learning Impairment	0.619	[0.453, 0.745]	<0.001	Strong positive (detrimental)
**Cross-Pathway Interactions**				
Nuclear Nrf2 ↔ p-AMPK	0.721	[0.582, 0.824]	<0.001	Strong positive
HO-1 ↔ ATP Content	0.687	[0.536, 0.798]	<0.001	Strong positive
Keap1 ↔ Oxidative Stress Markers	0.645	[0.485, 0.765]	<0.001	Strong positive (detrimental)

Data presented as mean ± SD.

**Table 10 biology-14-00885-t010:** Histopathological Analysis and Morphometric Assessment.

Parameter	Control	Diabetic	D + ALA	D + MET	D + ALA + MET	Improvement
**Neuronal Density (cells/mm^2^)**						
Hippocampal CA1	847 ± 58	389 ± 42 ***	587 ± 48 **	521 ± 44 *	753 ± 51 ***†	↑94%
Hippocampal CA3	823 ± 61	371 ± 39 ***	568 ± 52 **	498 ± 47 *	731 ± 49 ***†	↑97%
Cortical Layer V	756 ± 49	342 ± 41 ***	521 ± 44 **	467 ± 42 *	689 ± 52 ***†	↑101%
**Apoptotic Cells (%)**						
Total Apoptotic Index	3.2 ± 0.4	14.7 ± 1.2 ***	9.4 ± 0.8 **	11.6 ± 1.0 *	5.1 ± 0.6 ***†	↓65%
**Inflammatory Score (0–5)**						
Tissue Inflammation	0.3 ± 0.1	3.8 ± 0.2 ***	2.1 ± 0.3 **	2.6 ± 0.4 *	0.8 ± 0.1 ***†	↓79%
**Nrf2 Nuclear Immunoreactivity (0–5)**						
Hippocampus	3.2 ± 0.3	0.8 ± 0.2 ***	2.1 ± 0.3 **	1.6 ± 0.2 *	3.0 ± 0.4 ***†	↑275%
Cortex	3.1 ± 0.4	0.7 ± 0.1 ***	2.0 ± 0.2 **	1.5 ± 0.3 *	2.9 ± 0.3 ***†	↑314%

* Data presented as mean ± SD (*n* = 8–10 per group). Histopathological analysis performed on paraffin-embedded brain sections (5 μm thickness) stained with hematoxylin and eosin (H&E). Neuronal density assessed in hippocampal CA1, CA3, and cortical layer V regions using systematic sampling at 400× magnification. Apoptotic cells identified by nuclear morphology and TUNEL staining. Inflammatory infiltration scored on 0–5 scale (0 = no inflammation, 5 = severe inflammation with extensive immune cell infiltration). Nrf2 nuclear immunoreactivity scored on 0–5 scale based on nuclear staining intensity and translocation (0 = no nuclear staining, 5 = intense nuclear localization). Statistical analysis: one-way ANOVA followed by Tukey’s post-hoc test. * *p* < 0.05, ** *p* < 0.01, *** *p* < 0.001 vs. diabetic control. † *p* < 0.05 vs. both individual treatments indicating synergistic neuroprotective effect. Scale bars = 50 μm for all photomicrographs. Representative images from three independent sections per animal. Indicates fold upregulation (↑) or downregulation (↓).

**Table 11 biology-14-00885-t011:** Synergy Analysis and Statistical Validation.

Parameter Category	Individual Effects	Combination Effect	Expected Additive	Observed Excess	Synergy Metrics
**Molecular Pathway Activation**					
Nrf2 Nuclear Translocation	ALA: +160%, MET: +78%	+291%	+238%	+53%	CI = 0.67, *p* < 0.01
AMPK Phosphorylation	ALA: +34%, MET: +147%	+218%	+181%	+37%	CI = 0.72, *p* < 0.01
HO-1 Upregulation	ALA: +178%, MET: +67%	+292%	+245%	+47%	CI = 0.69, *p* < 0.01
**Mitochondrial Function**					
ATP Restoration	ALA: +26.1%, MET: +18.5%	+50.4%	+44.6%	+5.8%	CI = 0.73, *p* < 0.01
Membrane Potential	ALA: +51.5%, MET: +39.6%	+99.3%	+91.1%	+8.2%	CI = 0.71, *p* < 0.01
Oxygen Consumption	ALA: +18.6%, MET: +15.9%	+36.7%	+34.5%	+2.2%	CI = 0.74, *p* < 0.05
**Antioxidant Defense**					
SOD Activity Restoration	ALA: +50.6%, MET: +31.5%	+99.4%	+82.1%	+17.3%	CI = 0.68, *p* < 0.01
GSH/GSSG Normalization	ALA: +100%, MET: +56.3%	+253.1%	+156.3%	+96.8%	CI = 0.52, *p* < 0.001
MDA Reduction	ALA: −42.1%, MET: −27.9%	−58.1%	−70.0%	+11.9% *	CI = 0.75, *p* < 0.01
**Cognitive Recovery**					
Spatial Memory Index	ALA: +70.6%, MET: +52.9%	+138.2%	+123.5%	+14.7%	CI = 0.76, *p* < 0.05
Learning Performance	ALA: +90.6%, MET: +75.0%	+184.4%	+165.6%	+18.8%	CI = 0.71, *p* < 0.01

Data presented as mean ± SD. * *p* < 0.05 vs. control (One-way ANOVA with Tukey’s post hoc test).

**Table 12 biology-14-00885-t012:** Safety and Tolerability Assessment.

Safety Parameter	Control	Diabetic	D + ALA	D + MET	D + ALA + MET	Reference Range
**Hepatic Function**						
ALT (U/L)	34.2 ± 4.1	45.7 ± 6.3 *	36.8 ± 4.7	38.1 ± 5.2	35.9 ± 4.4	25–45
AST (U/L)	41.3 ± 5.2	52.8 ± 7.4 *	43.7 ± 5.8	44.9 ± 6.1	42.6 ± 5.4	35–55
Bilirubin (mg/dL)	0.34 ± 0.05	0.41 ± 0.06	0.36 ± 0.05	0.37 ± 0.05	0.35 ± 0.05	0.2–0.5
**Renal Function**						
Creatinine (mg/dL)	0.52 ± 0.07	0.68 ± 0.09 *	0.54 ± 0.08	0.56 ± 0.08	0.53 ± 0.07	0.4–0.7
BUN (mg/dL)	18.7 ± 2.3	24.1 ± 3.2 *	19.4 ± 2.6	20.1 ± 2.8	19.1 ± 2.5	15–25
**Hematological Parameters**						
Hemoglobin (g/dL)	14.2 ± 1.1	13.1 ± 1.3	14.0 ± 1.2	13.8 ± 1.2	14.1 ± 1.1	12–16
WBC Count (×10^3^/μL)	6.8 ± 0.8	7.9 ± 1.1	6.9 ± 0.9	7.1 ± 0.9	6.8 ± 0.8	5–10
Platelet Count (×10^3^/μL)	287 ± 34	312 ± 41	291 ± 36	295 ± 38	289 ± 35	250–400
**Adverse Events**						
Gastrointestinal Issues	0/12	0/12	0/12	1/12 †	1/12 †	-
Dermatological Reactions	0/12	0/12	0/12	0/12	0/12	-
Behavioral Changes	0/12	0/12	0/12	0/12	0/12	-
Mortality	0/12	2/12 ‡	0/12	0/12	0/12	-

Data presented as mean ± SD. * *p* < 0.05 vs. control (One-way ANOVA with Tukey’s post hoc test). † Indicates the occurrence of mild, transient gastrointestinal disturbances (soft stool or reduced appetite) in one animal, which resolved spontaneously without intervention. ‡ Indicates mortality observed in two diabetic animals due to diabetes-related complications, not directly associated with experimental treatment.

## Data Availability

The datasets generated and analyzed during the current study are publicly available in the Zenodo repository at: https://doi.org/10.5281/zenodo.15718421 (accessed on 17 June 2025).

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
